# PROBABLY APPROXIMATELY CORRECT LABELS

## ABSTRACT

Obtaining high-quality labeled datasets is often costly, requiring either human annotation or expensive experiments. In theory, powerful pre-trained AI models provide an opportunity to automatically label datasets and save costs. Unfortunately, these models come with no guarantees on their accuracy, making wholesale replacement of manual labeling impractical. In this work, we propose a method for leveraging pre-trained AI models to curate cost-effective and high-quality datasets. In particular, our approach results in *probably approximately correct labels*: with high probability, the overall labeling error is small. Our method is nonasymptotically valid under minimal assumptions on the dataset or the AI model being studied, and thus enables rigorous yet efficient dataset curation using modern AI models. We demonstrate the benefits of the methodology through text annotation with large language models, image labeling with pre-trained vision models, and protein folding analysis with AlphaFold.

## 1 INTRODUCTION

A key ingredient in any scientific pipeline is the availability of large amounts of high-quality *labeled* data. For example, social scientists rely on extensively-labeled datasets to understand human behavior (Salganik, 2017) and design policy interventions. Collecting high-quality labels for a given set of inputs is typically an arduous task that requires significant human expertise, costly large-scale experimentation, or expensive simulations. As such, researchers often outsource label collection to a third party "data provider"—this might be an annotation platform for labeling images, a wet lab for running scientific experiments, or a survey platform for collecting responses from a target population of individuals.

For data providers, the high cost of collecting high-quality labels combined with the rising performance of AI models suggests an enticing prospect: using AI *predictions* in place of manually-collected labels. Indeed, recent works have demonstrated AI models' ability to predict protein structures (Jumper et al., 2021), to evaluate language model responses (Zheng et al., 2023), and even to simulate human experimental subjects (Argyle et al., 2023). These advances highlight the potential for AI to streamline data annotation, and to produce high-quality labels at a fraction of the cost.

The problem with such an approach is that AI models are not always accurate, and come with no guarantees on how well they will label a given dataset. This makes it untenable to use AI-predicted labels as a direct substitute for expert labels, particularly in settings where label quality is critical. For instance, if the downstream goal is to draw conclusions that inform policy decisions, we should not blindly treat AI predictions of human behavior as if they were experimentally collected data.

Motivated by this state of affairs, in this paper we ask:

*Can we leverage powerful AI models to label data, while still guaranteeing quality?*

We answer this question in the affirmative, and provide a method—which we call *probably approximately correct* (PAC) labeling—that automatically combines cheap, non-expert labels (whether AI predictions, crowd-sourced labels, or simple heuristics) with expensive, expert labels to produce a labeled dataset with small error. PAC labeling yields guarantees similar in flavor to that of its namesake in probably approximately correct (PAC) learning (Valiant, 1984): given user-specified constants $\epsilon, \alpha > 0$, our procedure results in a labeled dataset with error at most $\epsilon$, with probability at least $1 - \alpha$. This guarantee is *nonasymptotic* under minimal assumptions on the dataset or the predicted labels being used.

## 1.1 CONTRIBUTIONS

We give a brief overview of our contributions, beginning with the problem setup. Given an unlabeled dataset $X_1, \ldots, X_n \in \mathcal{X}$, with unknown expert labels $Y_1, \ldots, Y_n$, our goal is to return a labeled dataset $(X_1, \tilde{Y}_1), \ldots, (X_n, \tilde{Y}_n)$, such that we incur only a small amount of labeling errors:

$$\frac{1}{n} \sum_{i=1}^{n} \ell(Y_i, \tilde{Y}_i) \leq \epsilon, \text{ with probability } 1 - \alpha. \tag{1}$$

Here, $\alpha$ and $\epsilon$ are user-chosen error parameters and $\ell$ is a relevant error metric. For example, if we want categorical labels to be accurate, we can choose the 0-1 loss: $\ell(Y_i, \tilde{Y}_i) = \mathbf{1}\{Y_i \neq \tilde{Y}_i\}$. The guarantee (1) then requires that at most an $\epsilon$-fraction of the dataset is mislabeled, with high probability. In regression problems, one might choose the squared loss, $\ell(Y_i, \tilde{Y}_i) = (Y_i - \tilde{Y}_i)^2$. We call $\tilde{Y}_i$ that satisfy the criterion (1) *probably approximately correct* (PAC) labels. To avoid strong assumptions, we treat the data as *fixed*; probabilities are taken only over the labeling algorithm.

To produce the label $\tilde{Y}_i$, we are allowed to query an expert for $Y_i$, which is costly, or instead use a cheap AI prediction $\hat{Y}_i = f(X_i)$, where $f$ is an AI model. The prediction $\hat{Y}_i$ can depend on any feature information available for point $i$, as well as any source of randomness internal to $f$. We will consider two settings: a basic setting with a single AI model $f$, and a more complex setting that assumes access to $k$ different models $f_1, \ldots, f_k$.

Of course, we can trivially achieve (1) by collecting expert labels for all $n$ data points. The goal is to achieve the criterion while minimizing the cost of the labeling. We will consider two ways of measuring the cost. The basic one is to simply count the number of collected expert labels; the AI-predicted labels are assumed to essentially come at no cost. The second way of measuring the cost takes into account the costs $c_1, \ldots, c_k$ of querying the $k$ models, as well as the cost of an expert label $c_{\text{expert}}$. When $c_{\text{expert}}$ is much larger than $c_1, \ldots, c_k$, the second setting reduces to the first.

Our main contribution is a method for producing PAC labels which, as we will show through a series of examples across data modalities and AI models, allow for significant saves in labeling cost. The key feature that enables a cost reduction is access to a measure of model uncertainty, which allows focusing the expert budget on instances where the model is most uncertain. Crucially, the nonasymptotic validity of PAC labeling does *not* depend on the quality of the uncertainties; more useful measures lead to larger saves in cost. We provide refinements of the method that additionally learn to calibrate the uncertainty scores to make the saves in cost even more pronounced.

## 1.2 RELATED WORK

**Adaptive dataset labeling and curation.** Our work most closely relates to the literature on efficient dataset labeling from possibly noisy labels. A distinguishing feature of our work is that we construct *provably accurate* labels with nonasymptotic guarantees, under no assumptions on the noisy labels. In contrast, much of existing work makes strong parametric or distributional assumptions—for example, model errors following a truncated power-law (Qiu et al., 2020) or a low-noise (Wang et al., 2021) distribution, the data following a well-specified parametric family (Ratner et al., 2016), or a class-conditional noise process (Northcutt et al., 2021). Many works lack formal accuracy guarantees (Zhu & Ghahramani, 2002; Iscen et al., 2019; Bernhardt et al., 2022; Li et al., 2023; Xie et al., 2020). Since we do not place distributional assumptions on the data but instead consider it fixed, our work particularly relates to the labeling problem known as transductive learning (Vapnik, 1998; Joachims, 2003). A key feature of our work is that we leverage pre-trained AI models, such as off-the-shelf language or vision models, and make no complexity assumptions on the expert labeling mechanism. An emerging line of work studies human-AI collaborative approaches to dataset curation (Li et al., 2023; Yuan et al., 2021; Liu et al., 2022; Kay et al., 2025). Our work is motivated by similar problems, with a focus on ensuring statistical validity. Importantly, many of the above works use uncertainty to decide which labels to collect (Bernhardt et al., 2022; Li et al., 2023; Kay et al., 2025). Our work similarly relies on uncertainty; in fact, our procedure can be applied as a wrapper around *any* uncertainty score to provide a statistically valid labeling. For example, the CoAnnotating paradigm defines an uncertainty score and proposes annotating the top $k$ most uncertain points with human annotations and the rest with AI annotations, for some user-chosen $k$. Our procedure can be applied to select $k$ in a data-driven manner, so that the final labeling is $(1 - \epsilon)$-accurate with high

probability. A similar observation applies to the CODA framework (Kay et al., 2025): our work can se applied as a wrapper around the expected information gain (EIG) used in CODA. More distant but related is a vast line of work studying different strategies for reliable aggregation of multiple noisy labels (Karger et al., 2014; Cheng et al., 2022; Dawid & Skene, 1979; Whitehill et al., 2009; Zhang & Chaudhuri, 2015; Yan et al., 2010; Welinder et al., 2010; Sheng et al., 2008; Yan et al., 2011).

**Distribution-free uncertainty quantification.** At a technical level, our procedure resembles the construction of risk-controlling prediction sets (Bates et al., 2021) and performing risk-limiting audits (Waudby-Smith et al., 2021; Shekhar et al., 2023). Like the former, our procedure bounds a monotone loss function by tuning a one-dimensional threshold, though not for the purpose of predictive inference. Similarly to the latter, our procedure aims to collect sufficient expert labels so as to meet a pre-specified quality guarantee. Like all these methods, PAC labeling satisfies *nonasymptotic*, *distribution-free* statistical guarantees. To achieve this, we build on betting-based confidence intervals (Waudby-Smith & Ramdas, 2024; Orabona & Jun, 2023). Our proposal relates in spirit to prediction-powered inference (Angelopoulos et al., 2023a; Zrnic & Candès, 2024a; Angelopoulos et al., 2023b) and related control-variate approaches (Zhou et al.; Egami et al., 2023), where the goal is to improve the power of statistical inferences given a small amount of expert-labeled data, a large amount of unlabeled data, and a good predictive model. We do not focus on statistical inference per se; rather, we construct an accurately labeled dataset that can be used for any downstream task.

**Active learning and inference.** The idea behind our method is to collect expert labels where the AI model is most uncertain; in that sense, our method relates to active learning (Settles, 2009; Lewis, 1995; Beluch et al., 2018; Zhang & Chaudhuri, 2015) and active inference (Zrnic & Candès, 2024b; Gligorić et al., 2024). Notably, there is a line of work in active learning that considers costs (Settles et al., 2008; Donmez & Carbonell, 2008; Wang et al., 2016). Our goal is fundamentally different: it is neither fitting a predictive model nor statistical inference, but producing high-quality labeled data with a provable nonasymptotic guarantee under minimal assumptions. In general, this is neither necessary nor sufficient for active learning. A related problem is "learning to defer" (Madras et al., 2018), a framework for jointly optimizing a predictive model and decisions to defer to an expert. Learning to defer does not come with guarantees on the statistical validity of the final labels, and it does not assume black-box access to predicted labels; the learner trains a model from scratch.

## 2 PAC LABELING: CORE METHOD

We begin with the basic setting with one AI model that produces cheap labels. Thus, we have $\hat{Y}_i = f(X_i)$ for all data points. In addition, we assume access to scalar uncertainty scores $U_1, \ldots, U_n$ (typically scaled such that $U_i \in [0,1]$) corresponding to the predictions $\hat{Y}_1, \ldots, \hat{Y}_n$. We place no assumptions on the quality of $U_i$, however if lower $U_i$ correspond to more accurate predictions $\hat{Y}_i$, the procedure will achieve big gains. The PAC guarantee (1) holds no matter the quality of $U_i$.

The basic idea behind the procedure is to find an uncertainty threshold $\hat{u}$ and label all data points with uncertainty that exceeds this threshold, $U_i \geq \hat{u}$. The more accurate the predictions $\hat{Y}_i$ are, the higher this threshold will be. To explain how we set $\hat{u}$, we introduce some notation. Let $\ell^u(Y_i, \hat{Y}_i) = \ell(Y_i, \hat{Y}_i)\mathbf{1}\{U_i \leq u\}$ and $L^u = \frac{1}{n}\sum_{i=1}^n \ell^u(Y_i, \hat{Y}_i)$. Ideally, if we knew $L^u$ for every $u$, we would choose the *oracle threshold*:

$$u^* = \min\left\{U_i : L^{U_i} > \epsilon\right\}.$$

In other words, if we label all points with $U_i \geq u^*$, meaning $\tilde{Y}_i = Y_i\mathbf{1}\{U_i \geq u^*\} + \hat{Y}_i\mathbf{1}\{U_i < u^*\}$, then we satisfy $\frac{1}{n}\sum_{i=1}^n \ell(Y_i, \tilde{Y}_i) \leq \epsilon$ with probability one. The issue is that we do not have access to $Y_i$, and thus we cannot compute $L^{U_i}$. To resolve this issue, we estimate an upper bound on $L^{U_i}$ by initially collecting expert labels for a small subset of the data. We will soon explain such a strategy; for now assume that for every $\alpha \in (0,1)$ and every $u$, we can obtain a valid upper confidence bound on $L^u$ at level $1 - \alpha$, denoted $\hat{L}^u(\alpha)$: $\mathbb{P}(L^u \leq \hat{L}^u(\alpha)) \geq 1 - \alpha$. Note that we only require $\hat{L}^u(\alpha)$ to be valid one $u$ at a time, not simultaneously. Our empirical approximation of the oracle threshold is

$$\hat{u} = \min\{U_i : \hat{L}^{U_i}(\alpha) > \epsilon\}. \tag{2}$$

Therefore, we collect expert labels where our uncertainty is $\hat{u}$ or higher: $\tilde{Y}_i = Y_i\mathbf{1}\{U_i \geq \hat{u}\} + \hat{Y}_i\mathbf{1}\{U_i < \hat{u}\}$. We argue that such labels are PAC labels.

**Theorem 1.** *The labels* $\tilde{Y}_i = Y_i \mathbf{1}\{U_i \geq \hat{u}\} + \hat{Y}_i \mathbf{1}\{U_i < \hat{u}\}$, *with* $\hat{u}$ *given by* (2), *are PAC labels* (1).

Interestingly, notice that the proof only requires $\hat{L}^{U_i}(\alpha)$ to be valid *individually*, even though we form $n$ confidence bounds. This is a consequence of the monotonicity of $L^u$ in $u$, similar in spirit to how monotonicity enables the Dvoretzky–Kiefer–Wolfowitz inequality (Dvoretzky et al., 1956) and risk-controlling prediction sets (Bates et al., 2021) to be free of multiplicity corrections.

It remains to provide a method to compute $\hat{L}^{U_i}(\alpha)$. Given a hyperparameter $m$, we collect $m$ draws $\{i_1, \ldots, i_m\}$ independently as $i_j \sim \text{Unif}([n])$. Then, for all $j \in [m]$, we sample $\xi_{i_j} \sim \text{Bern}(\pi_{i_j})$, where $(\pi_1, \ldots, \pi_n)$ are arbitrary sampling weights, and collect $Y_{i_j}$ if $\xi_{i_j} = 1$. This results in a dataset of $m$ i.i.d. variables $\{\ell(Y_{i_j}, \hat{Y}_{i_j}) \frac{\xi_{i_j}}{\pi_{i_j}}\}_{j=1}^m$; therefore, we can estimate $\hat{L}^u(\alpha)$ as:

$$\hat{L}^u(\alpha) = \texttt{meanUB}(\{\ell(Y_{i_j}, \hat{Y}_{i_j}) \xi_{i_j}/\pi_{i_j} \mathbf{1}\{U_{i_j} \leq u\}\}_{j=1}^m; \alpha).$$

Here, $\texttt{meanUB}(\cdot; \alpha)$ is any method for computing a valid upper bound at level $1 - \alpha$ on the mean from an i.i.d. sample. Indeed, the samples $\ell(Y_{i_j}, \hat{Y}_{i_j}) \frac{\xi_{i_j}}{\pi_{i_j}} \mathbf{1}\{U_{i_j} \leq u\}$ are i.i.d. with mean $L^u$, since $\mathbb{E}[\xi_{i_j}/\pi_{i_j}|i_j] = 1$. The motivation for allowing adaptive sampling weights $\pi_i$ is to allow forming a tighter confidence bound through a careful choice of the weights, although even uniform weights $\pi_1 = \cdots = \pi_n = p \in (0, 1)$ are a reasonable choice in practice.

There are many possible choices for $\texttt{meanUB}(\cdot; \alpha)$: it can be a nonasymptotic procedure such as the betting-based confidence intervals (Waudby-Smith & Ramdas, 2024), or (if one is satisfied with asymptotic guarantees) simply a confidence bound based on the central limit theorem: $\texttt{meanUB}(\{\hat{Z}_j\}_{j=1}^m; \alpha) = \hat{\mu}_Z + z_{1-\alpha} \frac{\hat{\sigma}_Z}{\sqrt{m}}$, where $\hat{\mu}_Z$ and $\hat{\sigma}_Z$ are the empirical mean and standard deviation of $\{\hat{Z}_j\}_{j=1}^m$, respectively, and $z_{1-\alpha}$ is the $(1 - \alpha)$-quantile of the standard normal distribution. In our experiments, we will primarily focus on procedures with nonasymptotic validity.

We summarize the overall procedure in Algorithm 1 and its guarantee in Corollary 1.

**Corollary 1.** *For any valid mean upper bound subroutine* $\texttt{meanUB}$, *Algorithm 1 outputs PAC labels.*

**Uncertainty calibration.** The utility of PAC labeling crucially depends on the quality of the uncertainty scores. However, some data points $X_i$ might have more accurate uncertainties than others. For example, suppose we can partition the $X_i$'s into two groups: on one, the model is consistently overconfident, and on the other, the model is consistently underconfident. Then, PAC labeling will overcollect expert labels for the data points in the second group. In the extreme case, imagine the model is incorrect on data points from the first group but produces low uncertainties, and is correct on data points from the second group but produces high uncertainties. Then, all expert labels for the second group will be collected. This is clearly wasteful, especially if the second group is large.

We propose uncertainty calibration to mitigate this issue. A natural way to calibrate uncertainties arises when there is a collection $\mathcal{C}$ of (possibly overlapping) clusters in the data. These clusters could be implied by externally given features (e.g., demographics), or they could be discovered in a data-driven way. For the zero–one loss, we use the multicalibration algorithm from Hébert-Johnson et al. (2018), stated in Algorithm 2 in the Appendix for completeness, to learn the uncertainty adjustment for each cluster. In practice, we learn the adjustment by collecting expert labels for a small subset of size $m \ll n$ of the overall dataset and applying the correction to the remainder of the dataset.

---

**Algorithm 1** Probably Approximately Correct Labeling

**Input:** unlabeled data $X_1, \ldots, X_n$, predicted labels $\hat{Y}_1, \ldots, \hat{Y}_n$, uncertainties $U_1, \ldots, U_n$, labeling error $\epsilon$, error probability $\alpha \in (0, 1)$, sample size for estimation $m$, sampling weights $\pi_1, \ldots, \pi_n$
  1: Sample $i_j \sim \text{Unif}([n])$ and $\xi_{i_j} \sim \text{Bern}(\pi_{i_j})$ independently for $j \in [m]$
  2: Collect $Y_{i_j}$ if $\xi_{i_j} = 1$ for $j \in [m]$
  3: Compute confidence bound $\hat{L}^u(\alpha) = \texttt{meanUB}\left(\{\ell^u(Y_{i_j}, \hat{Y}_{i_j}) \frac{\xi_{i_j}}{\pi_{i_j}}\}_{j \in [m]}; \alpha\right), \forall u \in \{U_i\}_{i=1}^n$
  4: Let $\hat{u} = \min\{U_i : \hat{L}^{U_i}(\alpha) > \epsilon\}$
  5: Collect true labels $Y_i$ for points where $U_i \geq \hat{u}$
  6: Let $\tilde{Y}_i \leftarrow Y_i \mathbf{1}\{U_i \geq \hat{u}\} + \hat{Y}_i \mathbf{1}\{U_i < \hat{u}\}$ for all $i \in [n]$
  7: For all $\{i_j\}_{j \in [m]}$ s.t. $\xi_{i_j} = 1$, (possibly) update $\tilde{Y}_{i_j} \leftarrow Y_{i_j}$
**Output:** labeled dataset $(X_1, \tilde{Y}_1), \ldots, (X_n, \tilde{Y}_n)$

---

# 3 MULTI-MODEL LABELING VIA THE PAC ROUTER

In many cases, we have access to several different sources of non-expert predictions. For example, we might have labels from several different AI models, or from (non-expert) human annotators of varying skill levels. In such settings, we might hope to leverage the strengths of these different predictors to reduce our overall labeling cost.

Concretely, consider a setting with $k$ cheap labeling sources; for each data point $i$, each source $j \in [k]$ provides a predicted label $\hat{Y}_i^j$ and an uncertainty $U_i^j$. Our goal is to route each data point to the most reliable source, minimizing the number of expert labels that we need to collect to retain the guarantee (1). (We later move to a cost-sensitive setting.) Our high-level approach has two steps:

1. First, we will learn a *routing model* $w_\theta : \mathcal{X} \to \Delta^{k-1}$ that maps each data point to a distribution over the $k$ labeling sources. We use the routing model to find the best source $j_i^*$ for each data point $i$, to which we assign label $\hat{Y}_i = \hat{Y}_i^{j_i^*}$ and uncertainty $U_i = U_i^{j_i^*}$.
2. We then apply the PAC labeling procedure from Section 2 to the selected data points, using the routed labels and uncertainties.

The main question is how to learn the routing model $w_\theta$. Throughout, we will assume access to a small, fully labeled *routing dataset* of size $m$, for which we observe $(X_i, Y_i, \{\hat{Y}_i^j, U_i^j\}_{j=1}^k)_{i=1}^m$, which we can use to learn the routing model.

A natural first idea (but ultimately a suboptimal one) is to maximize the expected accuracy of the routed labels—i.e., to solve $\arg\min_\theta \sum_{i=1}^m \sum_{j=1}^k w_{\theta,j}(X_i)\ell(Y_i, \hat{Y}_i^j)$, where $w_{\theta,j}(X_i)$ denotes the $j$-th coordinate of $w_\theta(X_i)$. This router is suboptimal because it ignores models' uncertainties and our error tolerance $\epsilon$. For example, consider the case where one labeling source has 100% accuracy but is also highly uncertain. For the purposes of PAC labeling, this source is not helpful; indeed, it will result in more expert labels being collected than if we had used the other sources. The router, however, will be incentivized to route all points to this source to maximize expected accuracy.

Can we route points in a way that takes into account the ultimate cost of the labeling procedure? To start, observe that the actual expected cost incurred by using a particular routing model $w_\theta$ is

$$\sum_{i=1}^m \sum_{j=1}^k w_{\theta,j}(X_i)\mathbf{1}\{U_i^j \geq \hat{u}\}, \tag{3}$$

where $\hat{u}$ is the threshold set by the PAC labeling procedure. Ideally, we could minimize this quantity directly, e.g., using gradient descent. There are two barriers to doing so: first, (3) is non-differentiable due to the $\mathbf{1}\{\cdot\}$ term, and second, $\hat{u}$ implicitly depends on the routing model $w_\theta$ itself.

To circumvent these issues, we first replace the indicator $\mathbf{1}\{U_i^j > \hat{u}\}$ with a sigmoid $\sigma(U_i^j - \hat{u})$. We then consider the following differentiable relaxation of the PAC labeling scheme that allows us to take gradients of our final objective with respect to the parameters of the routing model. Concretely, we consider a labeling scheme based on a threshold $\tilde{u}$ computed in the following way. We can approximate the PAC labeling guarantee with a weaker guarantee of expected average error control, then our procedure for finding $\tilde{u}$ can be written as:

$$\tilde{u} \approx \min \left\{ u : \mathbb{E}_{X_i, Y_i, j \sim w_\theta(X_i)}[\ell(Y_i, \hat{Y}_i^j) \cdot \mathbf{1}\{U_i^j \leq u\}] > \epsilon \right\}, \tag{4}$$

where the expectation over $X_i, Y_i$ denotes the empirical average over the (fixed) data points $(X_i, Y_i)$. If we again replace the indicator $\mathbf{1}\{U_i^j \leq u\}$ with a sigmoid, then $\tilde{u}$ is the solution to the equation:

$$\mathbb{E}_{X_i, Y_i} \left[ \sum_{j=1}^k w_{\theta,j}(X_i) \cdot \ell(Y_i, \hat{Y}_i^j) \cdot \sigma(\tilde{u} - U_i^j) \right] = \epsilon. \tag{5}$$

By strict monotonicity of the sigmoid and positivity of the remaining terms, this solution is unique. Therefore, we can write it as $\tilde{u}(\theta)$, and use the implicit function theorem to compute the gradient of $\tilde{u}$ with respect to $\theta$. After a short derivation deferred to App. A.3, we get:

$$\nabla_\theta \tilde{u}(\theta) = -\frac{\mathbb{E}_{X_i, Y_i} \left[ \sum_{j=1}^k \nabla_\theta w_{\theta,j}(X_i) \cdot \ell(Y_i, \hat{Y}_i^j) \cdot \sigma(u(\theta) - U_i^j) \right]}{\mathbb{E}_{X_i, Y_i} \left[ \sum_{j=1}^k w_{\theta,j}(X_i) \cdot \ell(Y_i, \hat{Y}_i^j) \cdot \sigma(u(\theta) - U_i^j) \cdot (1 - \sigma(u(\theta) - U_i^j)) \right]}. \tag{6}$$

This gradient admits a more compact representation using a single expectation (see App. A.3). This suggests a natural algorithm for training $w_\theta$: we compute the "smooth threshold" $\tilde{u}(\theta)$ by solving (5) (e.g., via binary search), and take a gradient step on the objective

$$\sum_{i=1}^{n} \sum_{j=1}^{k} w_{\theta,j}(X_i) \cdot \ell(Y_i, \hat{Y}_i^j) \cdot \sigma(\tilde{u}(\theta) - U_i^j),$$

using the gradient (6) to backpropagate through the threshold computation; and finally we repeat the above two steps until convergence.

**Recalibrating uncertainties.** Even with a principled way to route data points to different models, in practice our performance will often be bottlenecked by the quality of the uncertainties $U_i^j$. In particular, if all of the models are uncalibrated on a given data point, then routing will not yield any benefit in terms of the number of expert labels collected. Furthermore, the uncertainty values do not reflect the fact that we have routed the data point to the source we expect to be most reliable. Motivated by these observations, we propose a procedure for simultaneously learning a routing model and a better uncertainty model. The main idea is exactly the same as before: we will define an uncertainty model $u_\gamma : \mathcal{X} \to [0,1]$ that maps a data point to a new uncertainty value. To train the uncertainty model, we will use the same smoothed threshold procedure as before, noting now that the threshold $\tilde{u} = \tilde{u}(\theta, \gamma)$ depends on both the parameters of the routing model and the parameters of the uncertainty model. Accordingly, we perform gradient descent to solve the optimization problem

$$\min_{\theta,\gamma} \sum_{i=1}^{m} \sum_{j=1}^{k} w_{\theta,j}(X_i) \cdot \ell(Y_i, \hat{Y}_i^j) \cdot \sigma(\tilde{u}(\theta,\gamma) - u_\gamma(X_i)),$$

using implicit gradients $\nabla_\theta \tilde{u}(\theta, \gamma)$ and $\nabla_\gamma \tilde{u}(\theta, \gamma)$ derived using a similar logic as before (App. A.3).

**Cost-sensitive PAC router.** So far, we have treated the $k$ cheap labeling sources as if they are free (or vanishingly cheap, compared to the cost of the expert labeler). In practice, however, we may want to take the cost of the labeling sources into account. For example, these different sources may represent running experiments with different numbers of crowd workers, or with public APIs that have different costs. Suppose each labeling source $j$ has a per-label cost $c_j$, and that the cost of the expert labeler is $c_{\text{expert}}$. To incorporate costs, we use the same idea as the previous sections, aiming to directly optimize the expected cost incurred by the procedure. Our expected cost becomes

$$\sum_{i=1}^{m} \mathbb{E}_{j \sim w_\theta(X_i)} \left[ c_j \cdot \mathbf{1}\{U_i^j < \hat{u}\} + c_{\text{expert}} \cdot \mathbf{1}\{U_i^j \geq \hat{u}\} \right],$$

where $\hat{u}$ is the threshold computed using the main PAC labeling procedure. Just as in the previous sections: we approximate this threshold with a smoothed threshold $\tilde{u}$; use implicit differentiation to derive the gradient of $\tilde{u}$ with respect to the parameters of the routing and uncertainty models; replace indicators with sigmoids to get a fully differentiable objective; and perform gradient descent.

## 4 EXPERIMENTS

We evaluate PAC labeling on a series of real datasets, spanning natural language, computer vision, and proteomics. We repeat each experiment 1000 times and report the mean and standard deviation of the save in budget, i.e., the percentage of data points that are *not* expert labeled. We also report the $(1-\alpha)$-quantile of the empirical error $\frac{1}{n} \sum_{i=1}^{n} \ell(Y_i, \tilde{Y}_i)$ (which is supposed to be upper bounded by $\epsilon$). We plot the budget save against the realized error for 50 of the 1000 trials. We fix $\alpha = 0.05$. Except where otherwise noted, we use the betting algorithm of Waudby-Smith & Ramdas (2024) to compute mean upper bounds, with analogous results for the CLT-based upper bound in App. C.2.

We begin with PAC labeling experiments with a single AI model; we consider both problems with discrete and continuous labels. In this setting, we evaluate the benefits of uncertainty calibration. Then we study PAC labeling with multiple models, considering both the "costless" setting in which all AI predictions are treated as equally expensive, as well as the cost-sensitive setting that allows setting a different cost per AI model.

**PAC labeling with a single model.** We begin with the single-model case. In addition to PAC labeling, we consider two baselines. The first is the "naive" baseline, which collects expert labels

| Dataset | Metric | Method | | | |
|---|---|---|---|---|---|
| | | PAC labeling | Naive ($U_i \geq 0.1$) | Naive ($U_i \geq 0.05$) | AI only |
| **Media bias** | Budget save (%) | $(13.79 \pm 3.38)\%$ | 17.76% | 8.35% | — |
| | Error | 4.10% | 2.95% | 1.10% | 37.72% |
| **Stance on global warming** | Budget save (%) | $(28.09 \pm 3.28)\%$ | 62.51% | 25.10% | — |
| | Error | 4.57% | 10.13% | 0.83% | 24.79% |
| **Misinformation** | Budget save (%) | $(18.12 \pm 4.93)\%$ | 50.44% | 2.65% | — |
| | Error | 3.80% | 7.07% | 0.10% | 18.62% |

Table 1: **PAC labeling text datasets with GPT-4o.** We set $\epsilon = 0.05$. PAC labeling meets the error criterion, the AI only baseline has a large error, and the fixed threshold baseline is sometimes valid and sometimes not. Even when it is valid, it can be conservative.

| Dataset | Metric | Method | | | |
|---|---|---|---|---|---|
| | | PAC labeling | Naive ($U_i \geq 0.1$) | Naive ($U_i \geq 0.05$) | AI only |
| **ImageNet** | Budget save (%) | $(59.64 \pm 1.49)\%$ | 60.28% | 52.79% | — |
| | Error | 4.73% | 3.15% | 2.00% | 21.69% |
| **ImageNet v2** | Budget save (%) | $(39.07 \pm 2.67)\%$ | 46.05% | 39.07% | — |
| | Error | 4.74% | 4.31% | 2.62% | 35.33% |

Table 2: **PAC labeling image datasets with ResNet-152.** We set $\epsilon = 0.05$. PAC labeling and the fixed threshold baseline meet the error criterion and the AI only baseline has a large error. Even when it is valid, the fixed threshold baseline can be conservative.

for all points where the model's uncertainty is above a fixed threshold, such as $10\%$ or $5\%$. The second baseline is the method that only uses the AI labels, without using any expert labels.

*Discrete labels.* First we study the problem of collecting discrete labels; thus, we use the zero–one loss, $\ell(Y_i, \tilde{Y}_i) = \mathbf{1}\{Y_i \neq \tilde{Y}_i\}$. We consider several text annotation tasks from computational social science: labeling whether a text contains misinformation ($Y_i \in \{\texttt{misinfo}, \texttt{real}\}$) (Gabriel et al., 2022), labeling whether media headlines agree that global warming is a serious concern ($Y_i \in \{\texttt{agree}, \texttt{neutral}, \texttt{disagree}\}$) (Luo et al., 2020), and labeling of political bias of media articles ($Y_i \in \{\texttt{left}, \texttt{center}, \texttt{right}\}$) (Baly et al., 2020). We use predicted labels $\hat{Y}_i$ from GPT-4o, collected by Gligorić et al. (2024). For the uncertainties $U_i$, we use GPT's verbalized confidence scores; that is, we prompt the model to state its confidence in the answer. Additionally, we consider image labeling on ImageNet and ImageNet v2. We use the ResNet-152 to obtain $\hat{Y}_i$, and set $U_i = 1 - p_{\max}(X_i)$, where $p_{\max}(X_i)$ is the maximum softmax output given image $X_i$.

We summarize the results in Table 1, Table 2, and Figure 1. Using a fixed uncertainty threshold such as $5\%$ or $10\%$ results in highly variable results across datasets; sometimes the naive baseline is valid, sometimes it is not, and when it is valid often it is conservative. The approach of using AI labels alone achieves error that is far above the nominal. PAC labeling achieves error that fluctuates tightly around $\epsilon$, and the budget saves range between $14\%$ and $60\%$ depending on the difficulty of the labeling. We include plots analogous to those in Figure 1 for the remaining datasets in App. C.1.

*Continuous labels.* By choosing the appropriate loss, PAC labeling applies to continuous labels. The first task we consider is sentiment analysis (Socher et al., 2013). The goal is to provide a real-valued sentiment score $Y_i \in [0, 1]$ of a phrase, higher indicating more a positive sentiment. We use the squared loss, $\ell(Y_i, \tilde{Y}_i) = (Y_i - \tilde{Y}_i)^2$. We use GPT-4o to collect predicted labels $\hat{Y}_i$ and uncertainties $U_i$: we prompt GPT to predict an interval $[a_i, b_i]$ for the label $Y_i$, set $\hat{Y}_i = \frac{a_i + b_i}{2}$ and use the length of the interval as the uncertainty score, $U_i = b_i - a_i$. The second task is protein structure prediction. Here, $Y_i$ are experimentally derived structures and $\hat{Y}_i$ are AlphaFold predictions (Jumper et al., 2021). We use the mean squared deviation (MSD), the standard measure of protein structure quality, as the loss $\ell$. For context, two experimental structures for the *same* protein have a gap of around $0.36$ in terms of MSD. For the uncertainties $U_i$, we use the average predicted local distance difference test (pLDDT), AlphaFold's internal measure of local confidence. We use the CLT upper bound as the mean upper bound subroutine in the algorithm.

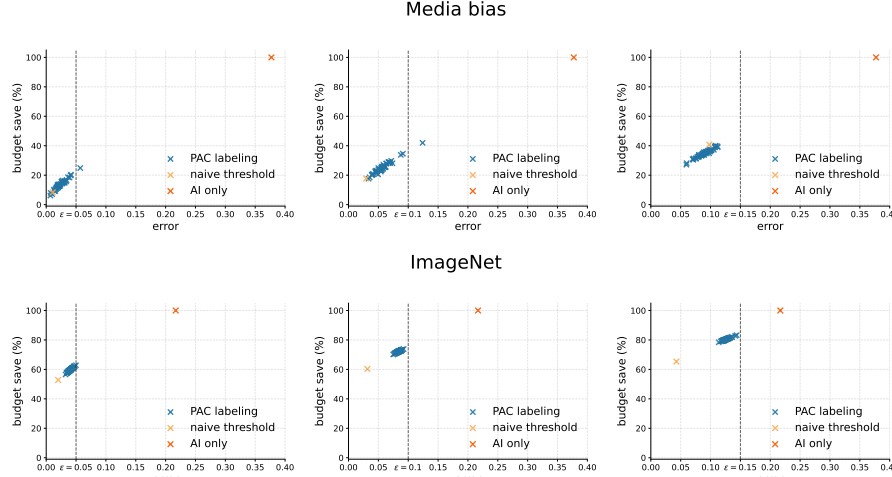

Figure 1: **PAC labeling for discrete labels.** Realized error and save in budget for PAC labeling, the naive thresholding baseline, and the AI only baseline. Each row and column correspond to a different dataset and value of $\epsilon$ (denoted by vertical dashed line), respectively. For the naive thresholding baseline, we collect expert labels for all points with $U_i \geq \epsilon$.

| Dataset | Metric | Method | | | |
|---|---|---|---|---|---|
| | | PAC ($\epsilon = 0.005$) | PAC ($\epsilon = 0.01$) | PAC ($\epsilon = 0.015$) | AI only |
| **Sentiment analysis** | Budget save (%) | $(16.03 \pm 2.49)\%$ | $(33.25 \pm 3.47)\%$ | $(50.86 \pm 3.93)\%$ | — |
| | Error | 0.004 | 0.009 | 0.013 | 0.021 |
| | | PAC ($\epsilon = 0.36$) | PAC ($\epsilon = 0.64$) | PAC ($\epsilon = 1.0$) | AI only |
| **Protein folding** | Budget save (%) | $(19.93 \pm 1.54)\%$ | $(26.47 \pm 3.37)\%$ | $(33.99 \pm 3.76)\%$ | — |
| | Error | 0.367 | 0.608 | 0.944 | 3.58 |

Table 3: **PAC labeling for continuous labels.** PAC labeling (approximately) meets the error criterion, while the AI only baseline has a large error.

| Dataset | Metric | Method | |
|---|---|---|---|
| | | PAC (before calibration) | PAC (after calibration) |
| **Media bias** | Budget save (%) | $(13.68 \pm 3.19)\%$ | $(16.72 \pm 2.81)\%$ |
| | Error | 4.10% | 4.22% |

Table 4: **Uncertainty calibration.** We set $\epsilon = 0.05$. PAC labeling with calibrated uncertainties (right) leads to higher saves than PAC labeling without calibration (left).

We summarize the results in Table 3 and Figure 2. For all $\epsilon$, PAC labeling tightly controls error while saving a nontrivial fraction of expert labels; the AI-only baseline does not meet the error criterion.

*Uncertainty calibration.* Calibrating uncertainties is a simple way to improve the performance of PAC labeling. In Table 4, we show the results of PAC labeling with GPT-4o on the media bias dataset (Baly et al., 2020), with a very simple calibration procedure: we use GPT-4o to cluster the articles into five clusters based on how conservative/liberal their source (e.g., CNN, Fox News, NYT, etc.) is, and we treat each article's cluster assignment as a group label $G_i$. We iterate through each group and uncertainty bin and additively adjust the uncertainties to match the average correctness using a small calibration set, as described in Section 2. Even in this simple setting (where the group labels are disjoint and derived only from the article source), calibration leads to a noticeable gain.

**PAC labeling with multiple models.** Next, we consider the multi-model case. We revisit the problem of annotating the political bias of media articles (Baly et al., 2020). In addition to GPT-4o predictions and confidences, we also collect predictions and confidences from Claude 3.7 Sonnet. We train a PAC router to route the articles between the two language models, while simultaneously training an uncertainty model, as described in Section 3.

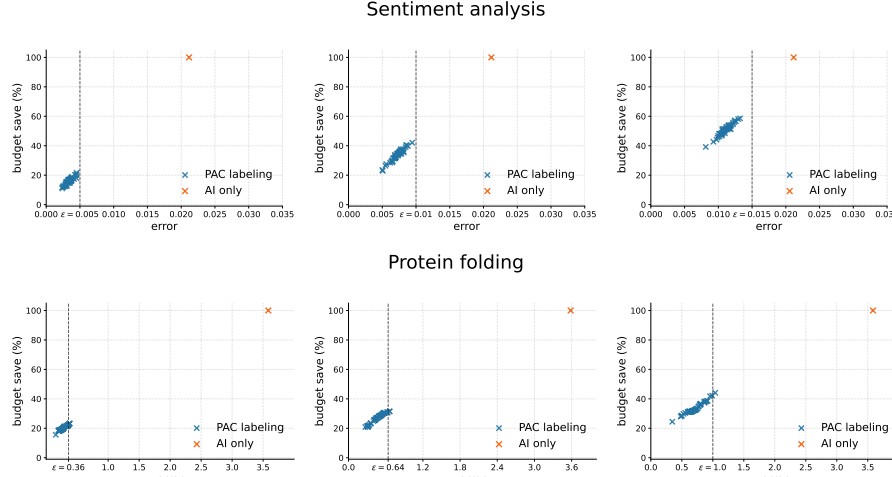

Figure 2: **PAC labeling for continuous labels.** Realized error and save in budget for PAC labeling and the AI only baseline. Each row and column correspond to a different dataset and value of $\epsilon$ (denoted by vertical dashed line), respectively.

| Dataset | Metric | Method | | |
|---------|--------|--------|--------|--------|
| | | PAC labeling (GPT-4o) | PAC labeling (Claude Sonnet) | PAC router |
| **Media bias** | Budget save (%) | $(13.79 \pm 3.38)\%$ | $(8.41 \pm 3.01)\%$ | $(41.61 \pm 1.50)\%$ |
| | Error | $4.10\%$ | $4.00\%$ | $4.61\%$ |
| | Save in cost | $(188.66 \pm 41.15)\%$ | $(131.36 \pm 49.20)\%$ | $(482.04 \pm 114.73)\%$ |
| | Error | $4.06\%$ | $3.58\%$ | $3.61\%$ |

Table 5: **PAC router for language models.** We set $\epsilon = 0.05$. The PAC router significantly improves the budget save (top) and save in cost (bottom) compared to PAC labeling with individual models.

*Costless predictions.* First we consider the setting of costless predictions, aiming only to minimize the number of collected expert labels. See Figure 3 (top) and Table 5 (top) for the results. GPT and Claude alone yield a $14\%$ and $8\%$ budget save, respectively, while by routing between the two saves about $42\%$ of the expert label cost. To give intuition for how this gain is achieved, in Appendix Figure 4 we plot the loss $L^u = \frac{1}{n}\sum_{i=1}^{n} \ell^u(Y_i, \hat{Y}_i)$ resulting from collecting labels at uncertainties greater than or equal to $u$, as a function of $u$. We observe that the router produces a curve $L^u$ that strictly dominates the loss curves of the individual models. This means that for any uncertainty threshold, the resulting labeling achieves a strictly smaller error than with a single model.

*Incorporating costs.* We also consider the cost-sensitive setting, where we take into account the costs of GPT-4o and Claude 3.7 Sonnet, and aim to minimize the overall labeling cost. We use the true current relative costs of the two models, setting $c_{\text{expert}} = 1$, $c_{\text{GPT}} = 0.25$, and $c_{\text{Claude}} = 0.075$. We show the results in Figure 3 (bottom) and Table 5 (bottom): cost-sensitive routing more than doubles the save in cost compared to GPT and more than triples the save compared to Claude.

## 5 DISCUSSION

This paper introduced probably approximately correct labels, a new approach to cost-sensitive dataset labeling assisted by off-the-shelf AI models and supported by rigorous statistical guarantees. Several directions for further development remain.

First, the procedure relies on a few user-specified hyperparameters—most notably the sample size $m$ used to estimate the underlying loss and the choice of sampling weights. Although we outline sensible practical defaults, it would be valuable to develop optimal automatic methods for selecting these parameters.

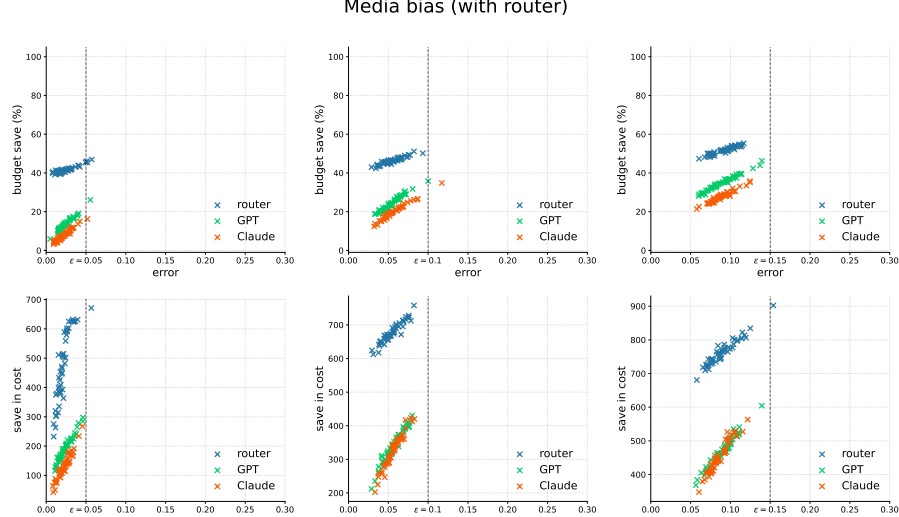

Figure 3: **PAC router for language models.** Realized error and save in budget for PAC labeling with GPT, PAC labeling with Claude, and the PAC router between GPT and Claude. The top row corresponds to the costless setting; the bottom row corresponds to the cost-sensitive setting. Each column corresponds to a different value of $\epsilon$ (denoted by vertical dashed line).

Second, while the PAC labeling guarantee itself is agnostic to the quality of uncertainty scores, our experiments show that good calibration is essential for achieving substantial budget saves. We considered both simple, multicalibration-style approaches, as well as black-box approaches to uncertainty calibration through the PAC router. One valuable direction for future work is to develop provably optimal uncertainty calibration techniques.

Finally, an implicit assumption made throughout was that the expert labels are correct; all validity claims are about agreement of the produced labels with expert labels. In practice, expert labels are inevitably imperfect. Extending PAC labeling to account for noisy or heterogeneous expert sources is an important direction for future work.

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

# A   DEFERRED DETAILS

## A.1   PROOF OF THEOREM 1

By the definition of $u^*$, we know $\frac{1}{n}\sum_{i=1}^{n}\ell(Y_i, \tilde{Y}_i) \leq \epsilon$ if $\tilde{Y}_i = Y_i\mathbf{1}\{U_i \geq u^*\} + \hat{Y}_i\mathbf{1}\{U_i < u^*\}$. Furthermore, by monotonicity, for any labeling threshold $u' \leq u^*$ the error criterion is satisfied. Therefore, on the event that $\hat{u} \leq u^*$, we know that $\frac{1}{n}\sum_{i=1}^{n}\ell(Y_i, \tilde{Y}_i) \leq \epsilon$.

We argue that $\mathbb{P}(\hat{u} \leq u^*) \geq 1 - \alpha$ as long as $\hat{L}^{U_i}(\alpha)$ are valid upper confidence bounds for all $U_i$. Suppose not: suppose $\hat{u} > u^*$. By definition, this must mean that $\hat{L}^{u^*}(\alpha) \leq \epsilon$. But at the same time, we know $L^{u^*} > \epsilon$; therefore, it must be that $\hat{L}^{u^*}(\alpha) < L^{u^*}$. This event happens with probability at most $\alpha$ because $\hat{L}^{u^*}(\alpha)$ is a valid upper confidence bound, and thus we have shown $\mathbb{P}(\hat{u} \leq u^*) \geq 1 - \alpha$.

## A.2   UNCERTAINTY MULTICALIBRATION

Below we state the algorithm for calibrating uncertainty scores, building on the multicalibration algorithm by Hébert-Johnson et al. (2018).

---

**Algorithm 2** Uncertainty multicalibration (Hébert-Johnson et al., 2018)

---

**Input:** uncertainties $U_1, \ldots, U_m \in [0, 1]$, expert labels $Y_1, \ldots, Y_m$, predicted labels $\hat{Y}_1, \ldots, \hat{Y}_m$, clusters $\mathcal{C}$, number of bins $B$, tolerance $\tau > 0$
1: Define bins $b_j = \left[\frac{j-1}{B}, \frac{j}{B}\right)$ for $j = 1, \ldots, B$
2: **repeat**
3:    updated $\leftarrow$ **False**
4:    **for** each cluster $C \in \mathcal{C}$ and each bin $j = 1, \ldots, B$ **do**
5:        Let $\mathcal{I}^{C,j} = \{i \in C : U_i \in b_j\}$
6:        **if** $|\mathcal{I}^{C,j}| > 0$ **then**
7:            Compute correction: $\Delta_{C,j} \leftarrow \frac{1}{|\mathcal{I}^{C,j}|}\sum_{i \in \mathcal{I}^{C,j}}\left(\mathbf{1}\{Y_i \neq \hat{Y}_i\} - U_i\right)$
8:            **if** $|\Delta_{C,j}| > \tau$ **then**
9:                Update: $U_i \leftarrow U_i + \Delta_{C,j}$ for all $i \in \mathcal{I}^{C,j}$
10:               updated $\leftarrow$ **True**
11: **until** updated is **False**
**Output:** calibrated uncertainties $U_1, \ldots, U_m$

---

## A.3   PAC ROUTER: DETAILS

Differentiating both sides of equation (4), we get

$$0 = \nabla_\theta \epsilon = \mathbb{E}_{X_i, Y_i}\left[\nabla_\theta \sum_{j=1}^{k} w_{\theta,j}(X_i) \cdot \ell(Y_i, \hat{Y}_i^j) \cdot \sigma(\tilde{u}(\theta) - U_i^j)\right]$$

$$= \mathbb{E}_{X_i, Y_i}\left[\sum_{j=1}^{k} \nabla_\theta w_{\theta,j}(X_i) \cdot \ell(Y_i, \hat{Y}_i^j) \cdot \sigma(\tilde{u}(\theta) - U_i^j)\right.$$

$$\left. + w_{\theta,j}(X_i) \cdot \ell(Y_i, \hat{Y}_i^j) \cdot \sigma(\tilde{u}(\theta) - U_i^j) \cdot (1 - \sigma(\tilde{u}(\theta) - U_i^j)) \cdot \nabla_\theta \tilde{u}(\theta)\right].$$

Rearranging, we get:

$$\nabla_\theta \tilde{u}(\theta) = -\frac{\mathbb{E}_{X_i, Y_i}\left[\sum_{j=1}^{k} \nabla_\theta w_{\theta,j}(X_i) \cdot \ell(Y_i, \hat{Y}_i^j) \cdot \sigma(u(\theta) - U_i^j)\right]}{\mathbb{E}_{X_i, Y_i}\left[\sum_{j=1}^{k} w_{\theta,j}(X_i) \cdot \ell(Y_i, \hat{Y}_i^j) \cdot \sigma(u(\theta) - U_i^j) \cdot (1 - \sigma(u(\theta) - U_i^j))\right]}.$$

We can estimate the above gradient using a single expectation, by defining the probability distribution

$$\eta_\theta(X, Y, \hat{Y}, U, j) \propto p(X, Y) \cdot w_\theta(X)_j \cdot \ell(Y, \hat{Y}^j) \cdot \sigma(\tilde{u} - U^j) \cdot (1 - \sigma(\tilde{u} - U^j)),$$

where $p(X, Y)$ is the (fixed) empirical distribution of data points with corresponding labels, such that

$$\nabla_\theta \tilde{u}(\theta) = -\mathbb{E}_{X,Y,j \sim \eta_\theta(X,Y,\hat{Y},U,j)} \left[ \nabla_\theta \log w_\theta(X)_j \cdot \frac{1}{1 - \sigma(\tilde{u}(\theta) - U^j)} \right].$$

For the setting with learned uncertainties, following the same logic, we have implicit gradients:

$$\nabla_\theta \tilde{u}(\theta, \gamma) = -\mathbb{E}_{X,Y,j \sim \eta_\theta} \left[ \frac{\nabla_\theta \log w_{\theta,j}(X)}{1 - \sigma(\tilde{u}(\theta, \gamma) - U^j)} \right] \quad \text{and} \quad \nabla_\gamma \tilde{u}(\theta, \gamma) = \mathbb{E}_{X,Y,j \sim \eta_\theta} \left[ \nabla_\gamma u_{\gamma,j}(X) \right].$$

### A.4 Loss $L^u$ after PAC routing

We plot the loss $L^u = \frac{1}{n} \sum_{i=1}^{n} \ell^u(Y_i, \hat{Y}_i)$ that results from collecting labels at uncertainties greater than or equal to $u$, as a function of $u$ in the context of the PAC router application from Section 4. To account for the fact that the different baselines might gives uncertainties $U_i$ of different magnitudes, without loss of generality we first map the uncertainties to their respective rank in $\{1, \ldots, n\}$. We observe that the router produces a curve $L^u$ that strictly dominates the loss curves of the individual models. This means that, for any uncertainty threshold, the resulting labeling achieves a strictly smaller error than with a single model. As a result, the critical uncertainty at which $L^u$ crosses error $\epsilon$ is significantly larger.

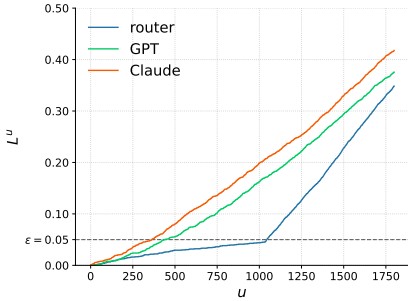

Figure 4: **Loss $L^u$ after PAC routing.** Error $L^u$ after collecting labels at uncertainties greater than or equal to $u$, as a function of $u$, for GPT and Claude individually and the PAC router. We observe that the router achieves a lower error $L^u$ than the individual baselines, for all $u$.

## B  Description of Protein Folding Experiment

Here, we briefly provide some background on the AlphaFold experiments performed in Section 4. We first queried the UniProt (The UniProt Consortium, 2025) database for reviewed human proteins for which 3D structures are available (`reviewed:true` and `structure_3d:true`). We then used the SIFTS database to map UniProt IDs to specific Protein Data Bank (PDB) chains. Predicted structures were retrieved from the AlphaFold Protein Structure Database (version 4) (Varadi et al., 2022), while corresponding experimental structures were downloaded from PDB (Berman et al., 2000). We then filtered and post-processed the chains and experimental structures using a basic structural alignment pipeline.

Using this pipeline, we constructed a dataset of 6668 proteins. For each sourced protein, our dataset contains (a) a sequence; (b) the experimental structure; (c) the predicted Local Distance Difference Test (pLDDT), a measure of confidence that is generated by AlphaFold[1], and (d) the Root Mean Square Deviation (RMSD), a measure of the ground-truth structural deviation between the predicted and observed structures[2]. Note that for proteins associated with multiple experimental structures,

---

[1]We refer the reader to `https://www.ebi.ac.uk/training/online/courses/alphafold/inputs-and-outputs/evaluating-alphafolds-predicted-structures-using-confidence-scores/plddt-understanding-local-confidence/` for further background on pLDDT.

[2]We refer the reader to `https://www.ebi.ac.uk/training/online/courses/alphafold/validation-and-impact/how-accurate-are-alphafold-structure-predictions/` for further background on RMSD.

we computed the RMSD against all available chains and selected the instance yielding the minimum RMSD to represent the optimal experimental baseline.

## C ADDITIONAL EXPERIMENTAL RESULTS

### C.1 DEFERRED PLOTS FROM THE MAIN TEXT

In Figure 5, we include plots analogous to those in Figure 1 for the remaining datasets: stance on global warming, misinformation, and ImageNet v2.

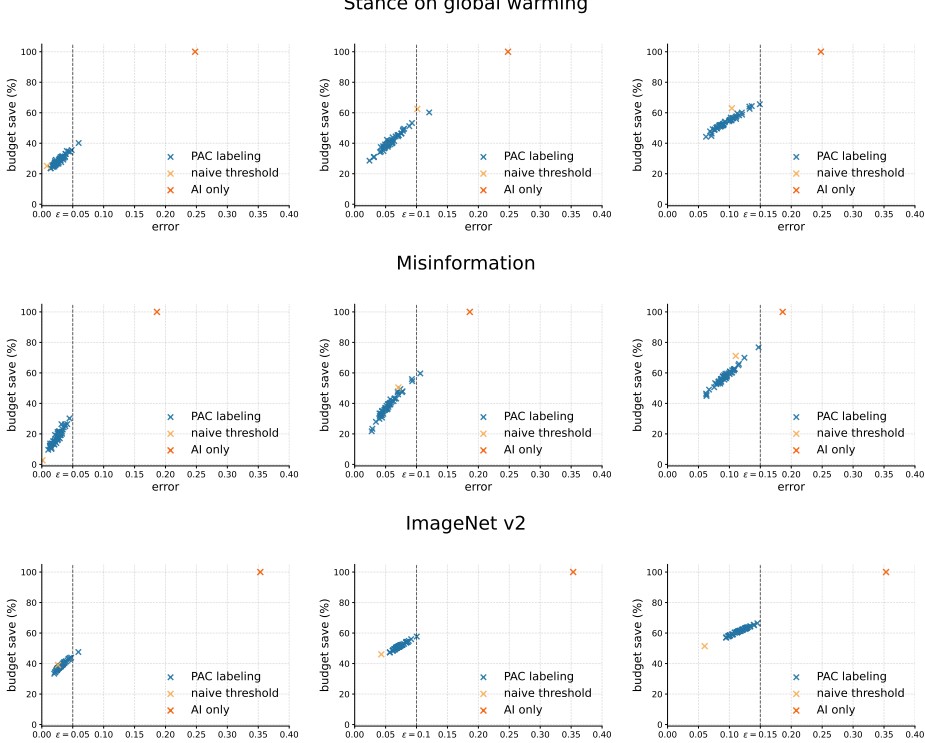

Figure 5: **PAC labeling for discrete labels (additional datasets).** Realized error and save in budget for PAC labeling, the naive thresholding baseline, and the AI only baseline. Each row and column correspond to a different dataset and value of $\epsilon$ (denoted by vertical dashed line), respectively. For PAC labeling, we plot the realized error and save in budget for $50$ randomly chosen trials. For the naive thresholding baseline, we collect expert labels for all points with $U_i \geq \epsilon$.

### C.2 EXPERIMENTS WITH ASYMPTOTIC CONFIDENCE INTERVALS

We include asymptotic analogues of the nonasymptotic results from Section 4. We rerun all experiments with discrete labels, this time using the asymptotic mean upper bound based on the central limit theorem (CLT) in the construction of PAC labels.

In Table 6 and Table 7 we compare PAC labeling with asymptotic and nonasymptotic guarantees on text and image datasets, respectively. We see that asymptotic confidence intervals, in addition to being easier to implement, enable larger budget saves compared to nonasymptotic intervals. The downside of relying on asymptotic guarantees is that the error rates might be slightly inflated—throughout we see error rates slightly above the nominal $5\%$.

In Figure 6 we show the realized budget save against the realized error when we use asymptotic intervals. Overall we see similar trends as in Figure 1, however the weaker requirement of asymptotic validity allows for generally larger saves.

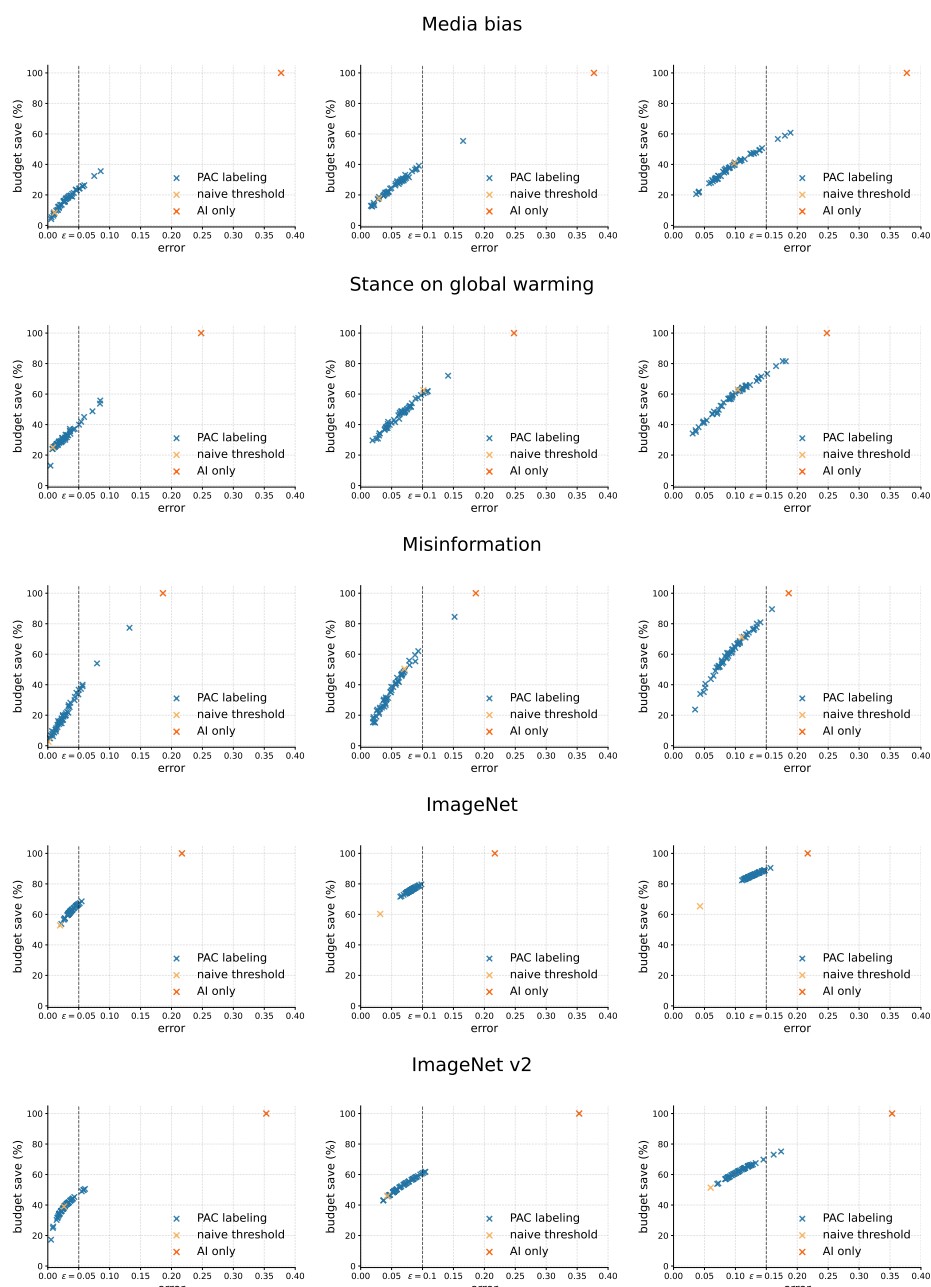

Figure 6: **PAC labeling for discrete labels with asymptotic confidence intervals.** Realized error and save in budget for PAC labeling, the naive thresholding baseline, and the AI only baseline. Each row and column correspond to a different dataset and value of $\epsilon$ (denoted by vertical dashed line), respectively. For PAC labeling, we plot the realized error and save in budget for $50$ randomly chosen trials. For the naive thresholding baseline, we collect expert labels for all points with $U_i \geq \epsilon$.

## C.3 EXPERIMENTS WITH VARYING $m$

Our PAC labeling algorithm (Algorithm 1) takes as input a sample size for estimation $m$. The choice of $m$ should consider two criteria: first, $m$ should be large enough such that estimating a one-dimensional mean with $m$ samples is accurate, and second, $m$ should be a relatively small fraction of the overall dataset size. The reason for the first criterion is that the procedure will be conservative, i.e. it will overcollect expert labels, if the upper confidence bound $\hat{L}^u(\alpha)$ is loose, which happens

| Dataset | Metric | Method | |
|---------|--------|--------|--|
| | | PAC labeling (asymptotic) | PAC labeling (nonasymptotic) |
| **Media bias** | Budget save (%) | $(16.11 \pm 6.96)\%$ | $(13.79 \pm 3.38)\%$ |
| | Error | 5.17% | 4.10% |
| **Stance on global warming** | Budget save (%) | $(32.15 \pm 7.38)\%$ | $(28.09 \pm 3.28)\%$ |
| | Error | 5.92% | 4.57% |
| **Misinformation** | Budget save (%) | $(21.41 \pm 10.95)\%$ | $(18.12 \pm 4.93)\%$ |
| | Error | 5.83% | 3.80% |

Table 6: **PAC labeling text datasets with GPT-4o, with asymptotic (left) and nonasymptotic (right) confidence intervals.** We set $\epsilon = 0.05$. PAC labeling with asymptotic guarantees enables larger saves, but may lead to slightly inflated error rates.

| Dataset | Metric | Method | |
|---------|--------|--------|--|
| | | PAC labeling (asymptotic) | PAC labeling (nonasymptotic) |
| **ImageNet** | Budget save (%) | $(62.82 \pm 2.57)\%$ | $(59.64 \pm 1.49)\%$ |
| | Error | 5.06% | 4.73% |
| **ImageNet v2** | Budget save (%) | $(39.20 \pm 5.82)\%$ | $(39.07 \pm 2.67)\%$ |
| | Error | 5.38% | 4.74% |

Table 7: **PAC labeling image datasets with ResNet-152, with asymptotic (left) and nonasymptotic (right) confidence intervals.** We set $\epsilon = 0.05$. PAC labeling with asymptotic guarantees enables larger saves, but may lead to slightly inflated error rates.

when $m$ is small. At the same time, $m$ being large means we are collecting many expert labels. Note that the two criteria do not depend on the specifics of the labeling problem much; to provide a heuristic that has robust performance for typical dataset sizes, we can take $m = \max\{500, 0.2n\}$.

In Figure 7, we plot the realized mean error and mean and standard deviation of the budget save for varying $m$ and $\epsilon = 0.05$. We see that the budget save peaks around 500 for smaller datasets, while for ImageNet the save keeps increasing for larger $m$ because $m = 2000$ is still negligible compared to the total dataset size.

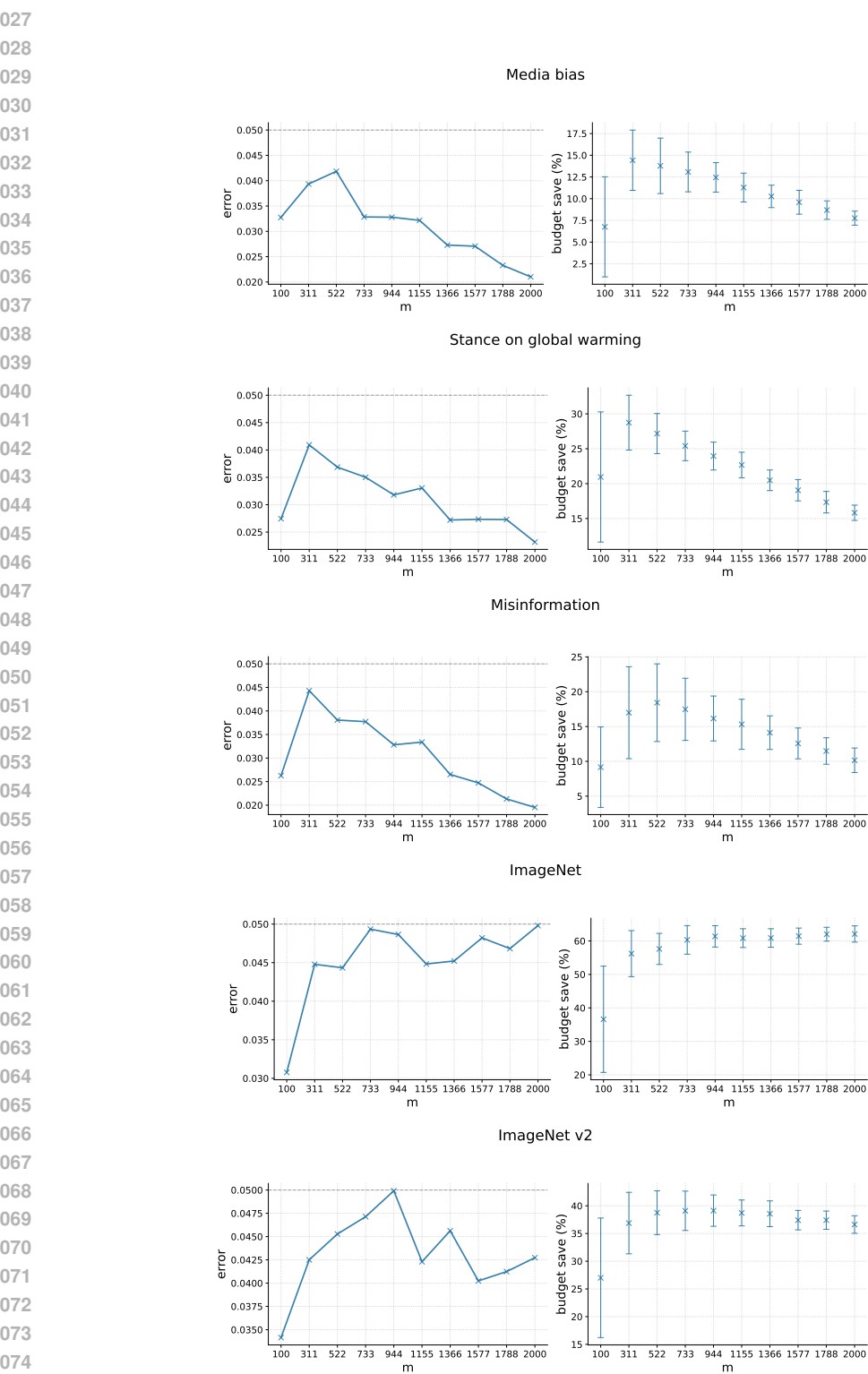

Figure 7: **Error and budget save for varying** $m$**.** Realized error and save in budget for PAC labeling with $\epsilon = 0.05$. Each row corresponds to a different dataset.

