# OpenReview forum: "Probably Approximately Correct Labels"
_ICLR.cc/2026/Conference — Submitted to ICLR 2026_

### Official Review · Reviewer_sfRp · 2025-10-24

**Soundness:** 3
**Presentation:** 2
**Contribution:** 2
**Rating:** 4
**Confidence:** 4

**Summary:**

This paper tackles the problem that building high-quality labeled datasets is costly, and while leveraging AI model predictions can reduce costs, the lack of accuracy guarantees makes full replacement difficult. It proposes a mathematical framework to label datasets using AI models, mathematically bounding the error below epsilon and the confidence above 1-a. Uncertainty scores play a central role in this process, where expert labels are collected only for data exceeding an uncertainty threshold u^. To ensure precise uncertainty measurement, calibration correcting uncertainty per data cluster is performed. The authors experimentally verify the broad effectiveness of the PAC approach across diverse domains including natural language, images, and proteins.

**Strengths:**

- Provides a formal mathematical framework to set an appropriate uncertainty threshold between expert labels and model-generated labels, with a rigorously defined iid variable within this framework.
- Clearly points out the critical importance of precise uncertainty measurement to guarantee the quality of AI-generated labels, enhancing uncertainty estimation with calibration techniques.
- Demonstrates through experiments on various data modalities—natural language, image, and protein structure—that the PAC method is a broadly applicable algorithm across multiple modalities and models.

**Weaknesses:**

- The uncertainty threshold is set universally across the dataset, but actual model inferences often show variations in uncertainty by class, due to reasons such as insufficient training samples for specific classes or confusing classes. Has the approach of using class-specific thresholds for more efficient labeling been considered?
- How is the value of m determined? This seems like a very critical hyperparameter for the method. If this value is heuristically determined, it would be difficult to consider the algorithm as robustly functioning. If it is described somewhere in the paper that I missed, it would be helpful to be informed.
- From the experimental results, it is hard to say that PAC brings a meaningful gain. In Figure 1, the PAC results lie between naive thresholding and AI-only interpolation, suggesting the PAC method mainly adjusts the trade-off between error and budget, rather than achieving gains beyond existing trade-off lines. If the authors’ intent is to enable trade-off adjustment using PAC, it makes sense, but this can partially be achieved just by constant threshold adjustment, limiting the contribution.
- There is no conclusion section, and therefore no mention of limitations and future work. Usually, limitations and future work part is considered as an important part of a paper, so this omission is a clear weakness.

**Questions:**

- As model overparameterization increases, over-confidence in model predictions worsens. Are there any countermeasures proposed for this?
- Depending on the reader’s background, some may find the format of protein folding labels unfamiliar. Would it improve the completeness of the paper to briefly explain the label format of the various domains you used in the experiment section?

---

> ### Author Response · Authors · 2025-11-21
> **Author Response**
>
> Thank you for your feedback and questions about our work! We address your concerns and questions one-by-one below.
>
> W1. We agree with the reviewer that there may be uncertainty variations for different classes. Our uncertainty calibration proposal (see line 183 in Section 2) aims to address precisely this problem, allowing for more general partitions of the data into groups beyond class membership. In our experiments in Table 4, we perform such calibration of uncertainties for different sources of media articles, which can be thought of as different classes.
>
> W2. We have added a new discussion of the choice of m with additional experiments where we vary m for five datasets that we consider in our experiments; thank you for raising this question. The choice of m is based on two criteria: (1) m should be large enough such that estimating a one-dimensional mean with m samples is accurate, and (2) m should be a relatively small fraction of the overall dataset size. The reason for (1) is that the procedure will be conservative, i.e. it will overcollect expert labels, if the upper confidence bound $\hat L^u(\alpha)$ is loose, which happens when m is small. At the same time, m being large means we are collecting many expert labels. Note that criteria (1) and (2) do not depend on the specifics of the labeling problem much; we can provide heuristics that have robust performance across all problems. For example, m=max(500, 0.2n) is a reasonable heuristic.
> In the new experiments (please see Figure 7 in the updated paper), we varied m between 100 and 2000. We see that the budget save peaks around 500 for smaller datasets, while for ImageNet the save keeps increasing for larger m because m=2000 is still negligible compared to the total dataset size.
>
> W3. The reviewer is correct that PAC labeling does not achieve a better tradeoff than the one defined by thresholding confidence scores, but this is actually by design — in fact, one can view the entire contribution of our method as finding the *right* confidence threshold in a statistically valid way, since this threshold will vary between different models and different datasets. Table 1 shows that setting this threshold arbitrarily leads to either loss of validity (if the threshold is set too high) or wasted budget (if the threshold is set too low). Setting the threshold in a naive data-dependent manner (e.g., by trying different thresholds and measuring the error and budget save on a heldout set) would also lose statistical validity because of test set reuse/dataset contamination—instead, our method offers a statistically principled way of saving on labeling costs while retaining the relevant guarantees.
>
> W4. Thank you for pointing this out. We have added a new conclusion section that includes a discussion of several limitations. Please see the new Section 5 in the revised version of the paper.
>
> Q1. Thank you for this great question - it turns out that by construction of our method, the budget saved by PAC labeling is only sensitive to the ordering, rather than the absolute value, of model confidences. Prior work (e.g. https://arxiv.org/abs/1706.04599) observes that while models get increasingly overconfident as they are overparameterized, the relative ranking of confidences is still indicative of model performance, which is why we still see significant budget gains even for large language models and deep neural networks.
>
> Q2. We thank the reviewer for this suggestion - in the revised version of the paper, please find an appendix (Appendix B) that provides more details on the protein folding experimental setup.
>
> Please let us know if you have further questions!

---

### Official Review · Reviewer_rfSa · 2025-10-29

**Soundness:** 3
**Presentation:** 3
**Contribution:** 2
**Rating:** 4
**Confidence:** 2

**Summary:**

This paper addresses the high cost of building high-quality labeled datasets. The authors claim that while many labeling methods exist, they have a critical limitation of providing no guarantees on accuracy. Therefore, the paper tackles the problem of how to guarantee the error rate, proposing a methodology called "PAC labeling." The method first uses AI labeling, classifies labels based on confidence, and then decides whether to use human labeling based on a certain threshold. The authors state this is done under minimal statistical assumptions. As a result, they achieve high performance compared to an "AI only" baseline and also show better performance than a fixed-threshold baseline.

**Strengths:**

This paper addresses an important problem and is well-timed with the current trend of synthetic data labeling, particularly in its focus on providing statistical guarantees.

**Weaknesses:**

The biggest problem is that the baseline comparison feels contrived, and the paper's contribution seems overstated. The authors claim that existing methods fail to provide mathematical guarantees, but such methodologies clearly exist. For example, Conformal Prediction (CP), which the authors themselves mention, deals with a very similar problem. The CP methodology quantifies prediction uncertainty into a prediction set with guaranteed coverage; it statistically guarantees the probability that the true label is within this set. Can this not be used to solve the same problem? The same applies to papers like "Learning to Defer...". This methodology also solves the same problem. Why did the authors not compare against these methods and evaluate their contributions toe-to-toe?

**Questions:**

See the Weakness section. Why were direct competitors that also provide statistical guarantees such as Conformal Prediction and Learning to Defer, omitted from the experimental comparison, and why was their contribution not evaluated "toe-to-toe"?

---

> ### Author Response · Authors · 2025-11-21
> **Author Response**
>
> Thank you for your comments and questions about our work! We address your main concern below. In short, *conformal prediction provides a fundamentally different guarantee from our work*.
>
> We agree with the reviewer’s characterization: conformal prediction (CP) outputs “a prediction set with guaranteed coverage; it statistically guarantees the probability that the true label is within this set.” Although we are familiar with this methodology and the surrounding literature, we do not see how to apply this method to output a dataset with PAC labels. We could apply CP to each data point $X_i$ individually, and we would have the guarantee that the prediction set $i$ contains the true label, but this does not imply anything about all labels $1,\dots,n$ being contained *simultaneously* in the produced prediction sets; for this, we would need to perform a Bonferroni correction. Furthermore, even if we performed such a correction (which would be extremely conservative), it remains unclear how to output a dataset with a single label per data point, as opposed to a set of labels per data point. Finally, there are certain additional technical conditions required in conformal prediction, namely exchangeability of the data points, which we do not require.
>
> Thank you for pointing us to the “learning to defer” work; we have added a discussion of it in our paper revision. Conceptually our work is indeed related to this framework: our method can be seen as learning when to use AI and when to “defer” to an expert. However, going beyond the conceptual connection, at a technical level the two are quite different. Learning to defer only considers binary labels, and their central proposal is to solve an optimization problem that trains a predictive model and deferral decisions together. The proposal does not come with guarantees on the validity of the final labels, and it does not assume black-box access to predicted labels. Our proposal has rigorous statistical guarantees, and we are motivated by settings where we already have good off-the-shelf predicted labels that we can leverage. Applying the learning-to-defer framework to our setting would require, for example, training an LLM and deferral decisions together, which is impractical.
>
> Please let us know if you have further questions!

---

### Official Review · Reviewer_R54A · 2025-10-29

**Soundness:** 2
**Presentation:** 3
**Contribution:** 2
**Rating:** 4
**Confidence:** 3

**Summary:**

The paper proposes a method to create high-quality labels by leveraging cheap predictions from AI models alongside costly expert labels. The method has statistical guarantee: with high probability, the final labeled dataset's error will not exceed a user-specified threshold. For data points labeled by AI models, only those with uncertainty exceeds a threshold are sent for expert labeling, minimizing overall cost. The paper also outlines a more complex PAC Router for optimally selecting between multiple AI models and presents experiments demonstrating budget savings.

**Strengths:**

- The paper considers a very practical setup with AI models generating labels for all data and then experts annotating the most valuable subset. This is helpful in generating labeled training data in low resource domains.
- The proposed method has nice statistical guarantee that with high probability, the final labeled dataset's error will not exceed a user-specified threshold.
- The effectiveness of the method in terms of cost saving is demonstrated empirically.

**Weaknesses:**

- In practice, the cost of high quality AI models is not neglectable, especially given the existence of test time scaling. Wonder how does this affect the proposed approach? Also wonder if allocating the whole expert budget to AI models test time scaling achieves better results?

- The considered baselines are too naive. A natural baseline is to use active learning. For example, first initialize the active learning model to be the model trained on labels generated by AI models, then select data points for experts to label using a standard active learning process. Also, a better baseline than the second baseline is to get more labels from AI models (with different prompts or AI models) until the cost matches the proposed method, and then aggregate the labels (e.g. with existing methods from the crowd sourcing literature).

**Questions:**

- how does the method compare to active learning or aggregated labels from many AI models (with different prompts and models).

---

> ### Author Response · Authors · 2025-11-21
> **Author Response**
>
> Thank you for your comments and questions about our work! We address each of them below. Please let us know if you have any further questions!
>
> W1. “In practice, the cost of high quality AI models is not neglectable…”: This is actually one of the main motivations behind our work, and can be captured neatly in the PAC routing framework (Section 3)! In particular, suppose one wants a principled way to decide, for each query, whether to send it to (a) a cheap, non-thinking model; (b) a more expensive reasoning model that exploits test-time scaling; or (c) a human expert, all while maintaining a desired level of accuracy. The PAC routing algorithm allows the user to specify the costs of each model and will automatically be able to minimize the cost (e.g., by using the non-thinking model on cases where it excels, the thinking model where it is necessary, and the human where both models are insufficient). Thus, we consider the scenario raised by the reviewer here to actually be a strength, rather than a weakness, of the PAC labeling framework.
>
> W2/Q1. This is an excellent set of questions, both effectively asking why PAC labeling is not compared to active learning or ensembling. To summarize, the main differences between the approaches are threefold:
> - First, the key property of our method is that it provides provable statistical guarantees on the error rate, which are crucial in high-stakes settings. Simple ensembling or active learning can often boost accuracy, but does not allow the user fine-grained control over the error rate. That said, many of these baselines operate by first constructing uncertainty scores (e.g., average model confidence, or disagreement between model predictions) and sampling based on the scores; note that *any* such method can be adapted in our framework (and thus translated into theoretical guarantees) by treating the computed scores as the uncertainty scores and running PAC labeling. In our work, we wanted to showcase the diversity of different data modalities (text, image, protein folding) and common uncertainty scores used in each, less so focusing on how to obtain the optimal uncertainty score.
> - Second, PAC labeling is designed to be a way to leverage black-box access to off-the-shelf (pre-trained) models like LLMs or AlphaFold, and so we do not assume that the user is able to train a model themselves. Many of the motivating examples that we experimented with are in, say, social science or biomedicine, fields where scientists are increasingly using AI as a black-box labeling tool, without knowing how to train such a model from scratch.
> - Finally, when routing between models we do not assume that we observe labels and uncertainty for all the models, but rather propose a method of optimizing a router directly - if the models are, e.g., different LLMs, the cost of ensembling their predictions will be the sum of their costs, but the cost of routing them will only amount to querying a single LLM for each example.

---

> ### Comment · Reviewer_R54A · 2025-11-25
>
> Thank you for answering my questions. While the responses have alleviated some of my concerns, I still believe the work would benefit from an empirical comparison with active learning or label aggregation methods:
> - First, there could also be guarantees on active learning[1] and label aggregation[2].
> - Second, while users might not able to train a model themselves, we could also just have a default active learning model or label aggregation model with fixed hyperparameters and with a very simple GUI for users to interact with, so that users do not need to be involved in model training details.
>
>
> [1]Wang, Zhilei, et al. "Neural active learning with performance guarantees." Advances in Neural Information Processing Systems 34 (2021): 7510-7521.
> [2]Ratner, Alexander J., et al. "Data programming: Creating large training sets, quickly." Advances in neural information processing systems 29 (2016).

---

> > ### Author Response · Authors · 2025-12-03
> > **Response**
> >
> > We appreciate the reviewer's response, and are very glad to hear that we addressed some of the concerns. Thank you for the followup and for the suggestions.
> >
> > 1. Regarding your first point: Although [1] and [2] come with guarantees, they make *very strong distributional assumptions*:
> >     - both papers assume binary classification,
> >     - paper [1] assumes the prediction algorithm is a particular fully connected neural network $f(x, \theta) = \sqrt{m} W_n \sigma (\cdots (\sigma (W_1 x))$ , and makes a strong Mammen-Tsybakov low noise condition,
> >     - paper [2] assumes that the distribution can be modeled by a *well-specified* parametric family and makes additional independence assumptions in the data.
> >
> >     Our claim is simple: we ensure a *provably non-asymptotically valid* labeling under *no distributional assumptions on the data* and *no assumptions on the quality or form of the AI-generated labels*. This is a far less restrictive setup than considered in [1] and [2]. We had previously discussed [2] in related work, and we have now also added [1] in the related work section.
> > 2. Regarding your second point: while the reviewer raises an interesting possibility, we believe that comparing our approach with such a method is out of the scope of our work. Training a model without exposing the user to the training process is often impractical and presents a separate set of challenges. Across many domains, practitioners use off-the-shelf models without any fine-tuning, since fine-tuning would require advanced knowledge; think of cases like protein folding where one would have to improve upon AlphaFold, or social scientists with no engineering background using large language models. Those practitioners may not have a machine learning expert at their disposal who can help them build the described GUI, nor do they have the knowledge to do this themselves.
> >
> > Thank you again for your comments and engagement with our work - we hope that the comments above address any remaining concerns.

---

### Official Review · Reviewer_FoeC · 2025-10-31

**Soundness:** 3
**Presentation:** 4
**Contribution:** 2
**Rating:** 4
**Confidence:** 3

**Summary:**

This paper introduces Probably Approximately Correct (PAC) Labeling, aiming to provide provable guarantees for AI-assisted labeling.
It proposes to select an uncertainty threshold such that all samples with model confidence above the threshold are labeled automatically, while others are sent to experts.
The guarantee states that, with high probability, the global labeling error is ≤ ε.
Extensions include a multi-model PAC Router and a practical multicalibration procedure to improve uncertainty reliability.

**Strengths:**

1. Relevant and well-motivated problem formulation. Applying PAC reasoning to automatic labeling is timely and practically meaningful.
   It formalizes a common heuristic (confidence-based filtering) into a statistically grounded process.

2. Simplicity and generality. The proposed approach is model-agnostic and directly applicable across diverse labeling pipelines.

3. Comprehensive empirical coverage. Experiments span multiple modalities and show consistent compliance with PAC bounds while reducing expert effort.

4. Clear presentation and awareness of practical issues.   The inclusion of multicalibration demonstrates attention to real-world miscalibration.

**Weaknesses:**

1. Limited overall contribution despite novel framing.
While the paper introduces new formulations (e.g., PAC Labeling and PAC Router) that are conceptually fresh, the underlying theoretical substance remains limited.
The analysis primarily builds on existing mean-upper-bound PAC results without introducing new bounds, assumptions, or insights into the nature of uncertainty in labeling.
Consequently, although the problem setting is well-motivated, the contribution lies more in repackaging and integration than in theoretical advancement.


2. Disconnect between calibration and PAC validity.
Multicalibration empirically adjusts uncertainty scores but may violate the monotonicity assumption required for PAC reasoning.
   The paper provides no formal analysis showing that PAC guarantees still hold after calibration.

3. Lack of intuitive and competitive comparisons.
Although experiments are broad, they do not clearly show cost–error trade-offs (e.g., comparing methods under equal error rates).
   Baselines are limited to simple heuristics; stronger competitors like conformal labeling or active selection are missing.
   As a result, the empirical advantage remains qualitative rather than quantitative.

**Questions:**

1. What concrete theoretical or methodological innovations are introduced beyond adapting existing PAC mean-upper-bound results?

2. Does multicalibration theoretically preserve the assumptions required for PAC guarantees, or is it purely an empirical fix?

3. Have the authors evaluated cost–error trade-offs, e.g., comparing expert costs at equal error levels across methods?

4. How does the framework behave under correlated errors or severe miscalibration beyond what multicalibration can correct?

---

> ### Author Response · Authors · 2025-11-21
> **Author Response**
>
> Thank you for your comments and questions about our work! We address them one-by-one below.
>
> W1. We are not aware of the “standard mean-upper-bound PAC results” that the reviewer is referring to, and would be happy to reference and more clearly compare to such work. As far as we know, *both the PAC labeling formulation and the PAC routing algorithm are novel contributions* that allow one to derive principled statistical guarantees about problems (co-annotation, model routing) that are typically heuristically decided. We would appreciate concrete references that we can discuss and compare our work to.
>
> W2. We believe that the reviewer may have misunderstood — *our paper does not make any monotonicity assumptions* —- the PAC labeling guarantee holds regardless of the uncertainty scores. The purpose of calibrating uncertainty scores is to enable larger saves in labeling cost; the core validity guarantee holds regardless. (The paper does discuss monotonicity of the loss function $L^u$, but that is not an assumption. It is guaranteed by construction.)
>
> W3. All our plots demonstrate a clear cost-error tradeoff: we evaluate the budget save (i.e. save in cost) against epsilon (the desired error budget). We disagree with the reviewer that the advantage of PAC labeling is “qualitative rather than quantitative” as demonstrated by the tables and figures in our work. *The error rates are strictly controlled under epsilon throughout*.
>
> *The conformal labeling (https://arxiv.org/pdf/2510.14581) paper appeared on arXiv after the ICLR submission deadline and thus we could not have compared to it.*
>
> Thank you for bringing up active selection (https://arxiv.org/pdf/2507.23771). We have included a discussion of this work in the revision of our paper. *However, this work comes with no theoretical guarantees and thus does not serve the same purpose as our work.* As we mentioned in line 102 in Section 1.2, our procedure can serve as a wrapper around any uncertainty score to provide statistical validity guarantees, and this also holds for the EIG score that defines active selection.
>
> Q1. See W1; we are not sure what “PAC mean-upper-bound results” refers to.
>
> Q2. Again, see W2; there is no such monotonicity assumption, and so multicalibration provably preserves the PAC labeling guarantee.
>
> Q3. See W3; our work does consider cost-error tradeoffs, and also varying expert costs in the PAC routing case.
>
> Q4. In general, the PAC labeling guarantee is *always* upheld, as it makes no assumptions whatsoever about the uncertainty scores. As the uncertainty scores become less correlated with true error, the budget save from PAC labeling decreases: for example, if the uncertainty scores are iid random samples from the unit interval, PAC labeling cannot achieve any budget save since there is nothing to be learned from the uncertainty scores.
>
> Please let us know if you have any further questions!

---

> > ### Comment · Reviewer_FoeC · 2025-11-27
> >
> > Thank you for your response. I now have a better understanding of the paper and see that there is good theoretical support.
> >
> > But I still have a few questions:
> >
> > - Please correct me if this is not right: in Figure 1, are the many PAC points in each subplot obtained by fixing $\alpha$ and $\epsilon$ then repeating the experiments multiple times?
> >
> > - My original concern in W3 was about the experiments. We need sufficiently convincing experiments to demonstrate the effectiveness of the method. In the paper, PAC was only compared with two very naive baselines: directly thresholding or using AI only. I just wonder if there are some other methods that could also be used to solve this problem. If you could compare with some of these methods, it might better demonstrate the effectiveness of PAC labeling.
> >
> > Additionally, I think it would help if there is a overview or outline for all experiments at the beginning of Sec. *Experiment*.

---

> > > ### Author Response · Authors · 2025-12-03
> > > **Response**
> > >
> > > Thank you for your reply - we are glad to hear that our response helped clarify the theoretical contribution of our work, and that the reviewer appreciates the paper and its theoretical novelty!
> > >
> > > Regarding your additional questions:
> > > 1. Yes, in Figure 1 we fix alpha and epsilon, repeat the procedure many times, and plot the results. We do this to illustrate the distribution of outcomes, since the procedure is inherently random.
> > > 2. We appreciate the reviewer’s question here: in our literature search, we did not find any methods (beyond the simple baselines we consider) which provided similar statistical guarantees under comparable assumptions. Furthermore, our method can be applied as a wrapper around any of those methods, ensuring *provably nonasymptically valid* labeling under *no distributional assumptions on the data* and *no assumptions on the quality of the AI-generated labels*.
> > >
> > > We have added an overview of the experiments at the beginning of the experimental section; please see the latest revision. Thank you for the suggestion.

---

### Meta-Review · Area_Chair_9CJV · 2025-12-22

**Summary:**

All the reviewers decide to reject this paper, this paper cannot be accepted in the current form.

**Reviewer Scores:**

NA

---

### Decision · Program_Chairs · 2026-01-26

Reject